# Cognition in Patients with Schizophrenia: Interplay between Working Memory, Disorganized Symptoms, Dissociation, and the Onset and Duration of Psychosis, as Well as Resistance to Treatment

**DOI:** 10.3390/biomedicines11123114

**Published:** 2023-11-22

**Authors:** Georgi Panov, Silvana Dyulgerova, Presyana Panova

**Affiliations:** 1Psychiatric Clinic, University Hospital for Active Treatment “Prof. Dr. Stoyan Kirkovich”, Trakia University, 6000 Stara Zagora, Bulgaria; 2Medical Faculty, University “Prof. Dr. Asen Zlatarov”, 8000 Burgas, Bulgaria; 3Medical Faculty, Trakia University, 6000 Stara Zagora, Bulgaria; presiana.panova@abv.bg

**Keywords:** schizophrenia, resistant schizophrenia, working memory, attention, fixation, reproduction, disorganized symptoms, dissociation, short-term memory, working memory

## Abstract

Schizophrenia is traditionally associated with the presence of psychotic symptoms. In addition to these, cognitive symptoms precede them and are present during the entire course of the schizophrenia process. The present study aims to establish the relationship between working memory (short-term memory and attention), the features of the clinical picture, and the course of the schizophrenic process, gender distribution and resistance to treatment. Methods: In total, 105 patients with schizophrenia were observed. Of these, 66 were women and 39 men. Clinical status was assessed using the Positive and Negative Syndrome Scale (PANSS), Brief Psychiatric Rating Scale (BPRS), Dimensional Obsessive–Compulsive Symptom Scale (DOCS), scale for dissociative experiences (DES) and Hamilton Depression Rating Scale (HAM-D)—cognitive functions using the Luria 10-word test with fixation assessment, reproduction and attention analysis. The clinical evaluation of resistance to the treatment showed that 45 patients were resistant to the ongoing medical treatment and the remaining 60 had an effect from the therapy. Results: Our study showed that, in most patients, we found disorders of working memory and attention. In 69.82% of the patients, we found problems with fixation; in 38.1%, problems with reproduction; and in 62.86%, attention disorders. Conducting a regression analysis showed that memory and attention disorders were mainly related to the highly disorganized symptoms scale, the duration of the schizophrenic process and the dissociation scale. It was found that there was a weaker but significant association between the age of onset of schizophrenia and negative symptoms. In the patients with resistant schizophrenia, much greater violations of the studied parameters working memory and attention were found compared to the patients with an effect from the treatment. Conclusion: Impairments in working memory and attention are severely affected in the majority of patients with schizophrenia. Their involvement is most significant in patients with resistance to therapy. Factors associated with the highest degree of memory and attention impairment were disorganized symptoms, duration of schizophrenia, dissociative symptoms and, to a lesser extent, onset of illness. This analysis gives us the right to consider that the early and systematic analysis of cognition is a reliable marker for tracking both clinical dynamics and the effect of treatment.

## 1. Introduction

The classical understanding of schizophrenia considers it a chronic mental illness with a varied clinical presentation and unclear etiology. The clinical presentation is related both to the presence of psychotic (delusions and hallucinations) and negative symptoms, but also to the presence of other clinical phenomena [1,2,3]. Apart from purely mental symptoms associated with abnormal changes in neuronal networks [4,5,6,7], inflammatory disorders associated with metabolic ones are also present in schizophrenia, as a cause and effect of the imbalance between the processes of neuroregeneration and neurodegeneration [8,9]. Traditionally, and in a therapeutic aspect, the dopamine hypothesis has been accepted, which tries to explain both the achieved therapeutic response during treatment with antipsychotic drugs and the development of possible refractoriness to the treatment [10,11]. Resistance as a clinical phenomenon is a challenge in all mental illnesses. Analysis of its prevalence shows a high percentage of 20–60% of patients [12]. No significant difference was observed in patients with schizophrenia. The big problem has always been how we define resistant cases. Over time, different criteria have been used to define resistant patients [13,14,15]. In search of a unified statement to be used in practice, a consensus statement was created to define resistant cases [16].

The median that runs longitudinally through the picture of schizophrenia drawn in this way is the cognitive disturbances that appear first in the course of the schizophrenic process and persist over time; at the end of the disease, their gradation to the development of demented symptoms is observed [17,18,19]. Some authors make an association between the development of cognitive symptoms in schizophrenia and frontotemporal dementia [17].

It was established that the cognitive deficit (which is entirely deducible) is the cause of the catastrophic psychosocial outcome of schizophrenia. Throughout the disease, a reduction in IQ from the norm of 100 to 70–85 has been reported [20,21,22]. Cognitive symptoms are also the primary tool for adaptation to the dynamically changing reality. In this sense, cognition is also the primary mechanism through which adaptation takes place [23,24]. Respectively and vice versa, their violation is the basis of their maladaptation and loss of social connectivity [25].

Cognitive deficits are the earliest and most socially significant symptoms of schizophrenia. They creep before the onset of the disease in the prodromal stage and are present at the debut of the disease [25,26]. They are a significant feature but, in clinical practice, they are not seen as core clinical symptoms, as we traditionally consider positive and negative symptoms [27,28]. Cognitive deficits worsen with the onset of the first psychotic episode, with research showing that they then return to baseline and remain relatively stable throughout the illness (of course, relative to the time of follow-up) [29,30]. When analyzing cognitive functions, we most often analyze the short-term memory and attention of the patient. The relationships between working memory (short-term memory in action) and attention are complex and overlapping [31,32,33].

Cognitive impairment is usually unresponsive to antipsychotic therapy. Therapeutic interventions have yielded conflicting results [28,34]. In addition to attempts at medication (at this stage without any particular result), various cognitive-rehabilitation techniques are used to improve cognitive functioning in patients with schizophrenia. Some authors have found positive and promising results using these methods [35], while others remain more skeptical about the results of their use [36]. A sizeable contemporary meta-analysis including 8851 patients shows that the use of cognitive remediation in patients with schizophrenia has a beneficial effect in terms of cognitive functioning [37,38]. These data give grounds for the European Psychiatric Association to make a consensus statement regarding the treatment of cognitive disorders in schizophrenia [39]. It is also necessary to take into account the fact that, in order to achieve recovery in patients with schizophrenia, it is necessary to bear in mind two main domains—clinical remission and social functioning [40]. Social functioning is also directly inferred from patients’ cognitive resources.

Data on the progression of cognitive impairment in schizophrenia are conflicting. Some studies show no cognitive deterioration over time, at least for the observed period (of one year) after the onset of the disorder. These observations lead some authors to believe that an underlying neurodegenerative process was not observed, as gradient changes in neurocognition were also not observed [41,42]. Other studies have also shown that no gradient deterioration is observed when conducting long-term longitudinal studies. There is evidence of improvement after the patient goes into clinical remission [43]. A nine-year follow-up finding confirmed the improvement in cognition during the first year found in other studies. The fact that the authors note is that, over the next five years, the curve of neurocognition flattens out and then falls again to be level at year 9 with the level of the first testing (before the improvement) [44]. Other researchers also concluded that relative stability in cognitive function was observed during the first ten years of treatment [45]. In the most extended study of cognition in patients with schizophrenia and hyperactivity disorder—13 years—it was concluded that, over time, there was a reduction in verbal memory and a general stagnation in attention and the speed of cognitive processes [46]. These studies show that, over a certain period, cognitive abilities in patients with schizophrenia recovered by reaching a certain level (that is, for a certain period they are a static phenomenon) and, after about 9–10 years, they again observed deterioration with a tendency to reach the initial values registered at the beginning of the disease [46]. Another meta-analysis showed that gradient morphological and neurocognitive changes were observed over time compared to an observed control group [47]. On the other hand, other researchers consider that it is necessary to take into account the presence of antipsychotic treatment, which can also contribute to the development of neuromorphological and neurocognitive disorders [48].

When we talk about cognition, we usually start using different terms, which do not lead to greater clarity. Given this fact, we stick to the main components that can be registered quickly in clinical practice—short-term memory and attention. The term working memory is also often used. However, it refers to short-term memory and its use in solving everyday problems, which also raises the question of participation in attention [49]. Working memory is a cornerstone in the schizophrenic process and its disorders are considered the most pronounced in these patients [50,51,52,53]. These and many other data give grounds for some authors to consider consciousness as a memory function developed from it in the course of evolution [54].

Analysis of attention shows that its disturbance is a fundamental cognitive deficit in patients with schizophrenia [55]. The authors consider that attention is an indicator that should be fundamental to measuring the condition and effectiveness of treatment in patients with schizophrenia. They found that attention was directly related to and inferred from the state of working memory [56]. In search of a practical approach to assessing cognition, the authors used the assessment of short-term memory and attention. Attention was assessed with memory curve analysis when conducting a memory assessment test [57,58].

A relationship between memory impairments and the degree of dissociation was found. Increased dissociation was associated with more significant memory impairments [59]. On the other hand, other authors have found that dissociation decreases with age in cognitively and non-cognitively impaired individuals [60].

These data gave us the basis to look for the relationships between memory and attention disorders in patients with treatment resistance, and, in those with a clinical effect, to establish as well the relationship with the course and clinical peculiarities of the gender distribution of the schizophrenic process in patients with schizophrenia.

## 2. Methods

### 2.1. Clinical Contingent

We analyzed 105 patients with schizophrenia. The gender distribution showed that 66 were female and 39 were male. The patients were admitted for treatment in a Psychiatric Clinic of the University Hospital in the city of Stara Zagora after the appearance of consecutive psychotic episodes. Patients were examined in the clinic’s outpatient practice and, after providing informed consent, were admitted for treatment and condition assessment. When analyzing their condition and inclusion in the study, inclusion and exclusion criteria were used. The patients were recruited and followed up from 2017 to 2022. The initial analysis and observation were carried out in hospital conditions, and, later, the observation and follow-up of their condition were carried out in outpatient conditions.

Inclusion criteria for patients with resistant schizophrenia [16] are as follows:Assessment of symptoms with the PANSS and BPRS scale [61,62].Prospective monitoring for at least 12 weeks.Administration of at least two antipsychotic medication trials at a dose corresponding to or greater than 600 mg chlorpromazine equivalents.Reduction of symptoms when assessed with the PANSS and BPRS scale by less than 20% for the observed period.The assessment of social dysfunction using the SOFAS scale is below 60.

Criteria for patients with schizophrenia in clinical remission are those who have met the criteria of the published consensus on remission in schizophrenia [63].

The exclusion criteria are as follows:Mental retardation.Psychoactive substance abuse.Presence of organic brain damage.Concomitant progressive neurological or severe somatic diseases.Expressed personality change (according to the diagnostic toolkit of DSM 5 and ICD 10). [64,65].Score of MMSE (Mini-Mental State Examination) below 25 points.Pregnancy and breastfeeding.

### 2.2. Assessment

Research has been used to assess cognitive impairment using the 10-word test [57,58]. This test is widely used and verified in many countries [66]. The methodology includes stimulus material, which the experimenter has developed himself. It is a set of 10 words, which should be common and short (from 1 to 2 syllables and should not be close in meaning). The two main memory processes are studied: (a) memorization (fixation) and (b) reproduction.

When experimenting, suitable conditions are necessary: A calm environment without interruption of the experiment, silence and a set of 10 words (monosyllabic or bisyllabic) that have no logical connection with each other. The instruction used by the experimenter consists of several stages: First explanation: “I will now read you ten words/words are read at intervals of one second in a clear voice. Listen carefully! When the reading finishes. Immediately repeat the words you have memorized. Their order does not matter. Got it?”. After the repetition, the experimenter places crosses under the reproduced words in the protocol. The instruction continues with a second explanation: “Now I will read the exact words to you again, and you must then repeat them. Furthermore, the ones told the first time and those missed altogether in whatever order wanted”. The experimenter again puts crosses under each repeated word. “One more time!” is required, and further repetitions of the set of words follow but without any instruction. If the subject says words that do not exist in the set, the experimenter records them by noting in which order these words are reproduced. If the person repeats the same word from the set several times in the protocol, many points are placed next to the cross as the number of times the examinee repeats this word. After five repetitions of the set of ten words, the experiment was terminated. Based on the data from the repetitions, an evaluation of the obtained memory curve is also made—unstable “zigzag”, plateau type, asthenic or average—giving information about the state of active attention [58].

Stimulus material (two examples):meat, glass, road, egg, birch, jam, goat, flag, sky, baghouse, horse, mushroom, honey, brother, forest, chair, bread, labor, oak

### 2.3. Statistical Analyses

Statistical package SPSS version 26 was used. The methods used were tailored to the specifics and objectives of the study. Non-parametric methods of analysis were used Mann–Whitney U test [67], correlation analysis and regression analysis). The regression analysis was conducted with the dependent variables being the studied fixation and reproduction. Independent variables were age, disease onset, duration of schizophrenia process, sex, weight, height, body mass index, PANSS positive, negative and disorganized symptoms, Dissociation Scale, Depression Scale and Obsessive–Compulsive Symptoms Scale. A correlation analysis was also conducted to find a relationship between the analyzed quantities.

The same cohort of patients was also investigated concerning other clinical characteristics such as depressive complaints, obsessive–compulsive and dissociative symptoms, lateralization of brain processes, the effect of the administration of the first antipsychotic medication and the gender-associated role in patients with schizophrenia [68,69,70,71,72,73,74,75].

All patients gave written informed consent before admission to the clinical settings and performing diagnostic tests and therapy. The study was conducted following the Declaration of Helsinki and approved by the Ethical Committee of University Hospital “Prof. Dr. Stoyan Kirkovich” Stara Zagora, protocol code TR3–02-242/30 December 2021.

## 3. Results

Of 105 patients, 45 have resistant schizophrenia, and the remaining 60 are in clinical remission. The gender distribution showed that 66 were women and 39 were men. The alignments according to the effects of treatment concerning age, onset of illness, duration of schizophrenia, BMI, height, education and handedness are presented in Table 1.

### 3.1. Assessment of Fixation

Among the 105 patients, we found fixation values within the normal range in 32 (30.18%). In the remaining 73 patients (69.82%), the fixation values were below this norm.

The mean fixation score for all patients we observed was 75.07, SD 15.561, with the minimum and maximum values being 45 and 100%, respectively. When analyzing the gender distribution, we found that, for females, the mean value of fixation was 77.33, with a standard deviation of 15.472. The minimum and maximum values on the scale were 48 and 100 points, respectively. Results for males showed a mean fixation value of 71.23. The standard deviation was 15.141, and the minimum and maximum values were 45 and 100, respectively.

The analysis of statistical significance showed a lack of statistical significance; the obtained value of “*p*“ is in a borderline range, *p* = 0.052 (Table 2, Figure 1).

When evaluating fixation in patients with resistant symptoms (RS), we found that the average value of fixation in them was 65.18%. The standard deviation was 11.161, and the minimum and maximum values were 45 and 100.

When evaluating the fixation in the patients in CR (clinical remission), we found an average fixation value of 82.48%. The standard deviation was 14.263, and the minimum and maximum values were 45 and 100, respectively.

From these results in the patients with R.S., it is evident that the fixation disorders in them as an average value can be described as moderately pronounced (although in the upper range of the assessment moderate). We see patients with lower fixation in the range of significant impairment and those within the normal range.

The data for patients in clinical remission show that the average value of 82.48 is at the upper end of mild disorders (near the lower end of the norm) (Application 1, Appendix A).

The comparison between the two groups showed the presence of a highly statistically significant difference (*p* < 0.001 ***) (Table 3, Figure 2).

In order to comprehensively analyze the influence of additional factors, such as the PANSS scales, body mass index, education, the onset of the disease, its duration, the age of the patients, Dissociation Rating Scale, Hamilton depression rating scale, Obsessive–Compulsive Symptoms Rating Scale, on fixation, we conducted a multiple regression analysis.

Three factors influenced fixation status significantly. Factors recorded were the PANSS—disorganized symptoms scale, the duration of the illness and the dissociative symptoms scale (Table 4).

Our data show that disorganized symptoms, duration of the schizophrenic process and high dissociation scale are the factors associated with a high degree of fixation involvement. There is also a statistically significant difference between patients with resistance to the treatment and those with an effect from it.

We looked for a relationship between the effect of fixation, the onset of the disease and the duration of the schizophrenic process by conducting a correlation analysis. We found a correlation between the fixation disorder, early onset of the disease, and the duration of the schizophrenic process. The results are presented in Table 5.

It is clear from the table that the duration of the schizophrenic process is a factor associated with a higher probability of fixation disorders, as both factors have clinically significant statistical significance in terms of their influence.

### 3.2. Reproduction

We observed 65 (61.90%) of the patients with reproduction values within the normal range. In the remaining 40 patients (38.1%), the reproduction values were below this norm.

The distribution by gender showed that the measured points on the reproduction scale for females was 88.44. The standard deviation was 16.471, with the minimum and maximum values being 50 and 100, respectively.

For males, the reproduction scale analysis showed a mean value of 83.56, with a standard deviation slightly higher than that of females—18.223. The minimum and maximum values were 45 and 100, respectively.

Statistical analysis showed no statistically significant difference (*p* > 0.05). In patients with resistant schizophrenia, the mean value on the reproduction scale was 77.07 and the standard deviation was 16.708, with a minimum value of 45 and a maximum of 100.

In responders to therapy, the mean value of the reproduction scale was 93.9, the standard deviation was 13.872, and the minimum and maximum values were 50 and 100. Statistical analysis using the Mann–Whitney U test for statistical dependence revealed a high correlation (Mann–Whitney U; 607.000; *p* < 0.001 ***).

In order to comprehensively analyze the influence of additional factors such as the PANSS scales, body mass index, education, the onset of the disease, its duration, the age of the patients, Dissociation Rating Scale, Hamilton depression rating scale and Obsessive–Compulsive Symptoms Rating Scale on reproduction, we conducted a multiple regression analysis (Table 6).

Analysis of the relationship between reproduction and the duration of schizophrenia showed the presence of a relatively weak correlative dependence. No correlation was found between reproduction and disease onset (Table 7).

### 3.3. Attention

The analysis of attention in the observed patients was carried out based on an outlined memory curve during the study to reproduce the results.

This analysis showed that, among the observed patients, 41 had an unstable (zigzag) memory curve, 17 had a plateau-type memory curve, 8 had an asthenic memory curve and 39 (37.14%) had a usually presented one.

In total, 66 (62.86%) were found to have an attention disorder. The distribution of the type of memory curve (attention) by gender is presented in Table 8 and Figure 3.

A statistical analysis showed that no statistically significant difference was observed in males and females regarding the type of memory curve and the involvement of attention in this process (*p* > 0.05).

When comparing the patients with schizophrenia according to the effect of the treatment, the following distribution was found (Table 9, Figure 4).

From the presented table and graph of the distribution of the type of attention engagement in these patients, it was found that, in the patients with resistance, more than half had an unstable type of memory curve, which is related to instability, i.e., inability to maintain active attention during the task at hand. Regression analysis assessing the influence of other factors such as height, weight, BMI, disease onset, duration, depression scale, dissociation scale, obsessive–compulsive symptoms, and the PANSS positive, negative and disorganized symptoms scales showed that disorganized symptoms and the onset of the disease are the factors influencing the attention disorders to the highest degree (Figure 5 and Figure 6, Table 10).

## 4. Discussion

Our research shows that, when assessing working memory and attention, there is no statistically significant difference between males and females. It is noteworthy that, in general, males perform slightly worse than females without reaching a statistically significant difference. These results, on the one hand, confirm the data of other authors on the lack of difference between males and females regarding the cognitive functions studied [76]. Other authors found a difference, with males showing more severe cognitive deficits than females [77]. Our results also show that males have lower scores on the cognitive test assessment without reaching a statistically significant difference to verify these differences. These differences may also be because, in our study, individuals of the female gender predominate, which is most likely also due to the inclusion and exclusion criteria we used. Other evidence also suggests that cognitive impairment is more common in males [78]. We found a significant difference in schizophrenia patients compared to the general population in terms of short-term memory—fixation and reproduction. Our results show that short-term memory impairments are more pronounced for fixation and less so for reproduction in patients with schizophrenia. Data from the literature indicate that cognitive impairment is generally observed in about 80% of patients with schizophrenia [79,80]. Our observation reflects only short-term memory and attention, as the disorders we found were of a lower percentage. In 69% of the patients, there is an impairment of fixation, in 38.1% an impairment of reproduction and, in 62.85%, a violation of attention. The difference in the percentages found (in our low-scoring study) is most likely due to the stricter exclusion criteria and the fact that we are limited to an analysis of short-term memory and attention. When comparing patients with resistance and those with a treatment effect, we found a significant difference in all measures, which was most pronounced in terms of fixation. Our observations confirm the results of other authors on the presence of differences between patients with and without deficits, and that deficits in memory and attention are associated with a poor prognosis [81,82]. The data from the literature show that the cognitive deficit is observed even before the onset of psychosis [83,84] and remains stable for a certain period [41,42,45], and after ten years a deterioration can be observed, usually associated with other factors such as BMI metabolic disorders as well as those involving other organs and systems [46]. The patients we analyzed from the two groups did not show differences in weight and BMI, so these factors cannot be assumed to influence cognitive performance [75].

The analysis of other researchers shows a different trend: cognitive symptoms develop in the first 10–20 years, after which their trajectory does not differ much from that of the main population [85]. These analyses, as well as our observations, pose the question of whether differences in cognition are not observed at the beginning of the disease, which predetermines the later resistance to treatment. Our observation is of patients with an average disease duration of 10.2 years. These observations of ours pose the question whether, on the one hand, the longer duration of the disease in patients with resistance (on average 14.31 years) is not also related to a re-started deterioration of cognition (observed after the 10th year) or, on the other hand, the difference in cognitive performance was present from the beginning of the disease in them?

We find an association of the cognitive impairments we measured with disorganized symptoms, which some authors find criteria for a high probability of resistance [86,87,88,89]. With our observations, we cannot support the results of other authors that all PANSS subscales are associated with cognitive impairment [90,91]. We report an association between cognitive symptoms and a high degree of dissociation. On the other hand, there is an established relationship between a high degree of dissociation and resistance in patients with schizophrenia [69]. We find cognitive impairments in terms of working memory and attention. What is necessary to note is the fact that, in more than half of the patients with resistance to the treatment, we find an unstable (zigzag type) memory curve, which indicates an instability or what we can express as a dissociation (fragmentation) of attention while completing the assigned task. We also established a relationship between the unstable type of memory curve and disorganized symptoms in patients with schizophrenia. We found that this relationship further provides an opportunity to elucidate the etiology of disorganized symptoms. The lack of active attention (established instability and fragmentation) is also associated with working memory disorders, which leads to disruption of the continuum in behavior and speech. These results shed light on the relationships between dissociation, resistance, disorganized symptoms and registered cognitive impairment. Regression analysis showed that attention disorders are also dependent on the onset of the disease. This observation of ours confirms the conclusion of other authors that attention is a factor associated with the severity of the schizophrenia process, and is an essential marker for the therapeutic dynamics and prognosis of the disease [55,56,92]. These results pose the question of searching for more complex therapeutic interventions in the treatment of patients [93].

On the other hand, we find impairments in working memory with an underlying fixation impairment. That gives us reason to disagree with the opinion of other authors about the possible similarity of dementia in patients with schizophrenia with frontotemporal dementia [17]. Memory impairment is not one of the cardinal symptoms of frontotemporal dementia or, if present, it occurs later in the nosological process [94,95]. On the other hand, both in schizophrenia and in other diseases, there are data on morphological changes in brain structures but they do not always correspond to the clinical picture, both in patients with schizophrenia and in some neurological diseases [96,97,98]. We find mainly problems with memory and attention. However, on the other hand, given that cognitive problems appear early before the onset of the acute psychotic state, the question of proximity to frontotemporal dementia can hardly be raised.

## 5. Limitations

The limitation in our study is also related to the need for more patients to make a complete verification of the obtained results. On the one hand, females predominate, which does not provide complete clarity on the gender differences concerning the identified cognitive disorders. On the other hand, we analyze short-term memory, and the question remains open whether these are also the early characteristics of cognitive disorders at the onset of psychosis and whether these fixation and reproduction disorders associated with a characteristic shape of the memory curve did not appear later in the course of schizophrenia process. Of course, our research shows that the duration of the schizophrenic process is a factor associated with cognitive disorders. However, this association does not answer the question of the extent to which differences in early cognitive impairment play a role in resistance to treatment. Thus, the question remains open as to whether the differences in cognitive impairment in patients with resistance and those in therapeutic remission were not present at the onset of psychosis or developed gradually during the progression of the psychotic state.

On the other hand, the question remains open as to whether we have chosen the most appropriate cognitive tool to analyze the examined patients. We use Luria’s 10-word test. This test is not very different from the tests created based on the ideas of George A. Miller with his magic number—7 [99]. Whether it is the best cognitive tool is a question that is difficult for anyone to answer, as is proposing the ideal and universal one. We use a cognitive tool that has been used in clinical practice in Bulgaria for almost 30 years to assess patients with schizophrenia, with which we have practice and experience.

## 6. Conclusions

We registered the presence of disorders in working memory—fixation and reproduction, as well as disorders in active attention in the studied patients with schizophrenia. These disorders are significantly more pronounced in patients with resistance to treatment. The cognitive symptoms we found were dependent on disease onset, duration of schizophrenia, high degree of dissociation and pronounced disorganized symptoms. Do not the data on a significantly greater disturbance of cognitive symptoms in patients with resistance give us the right to consider resistant schizophrenia as a more “organic” disease, in which the disturbance of cognition is the reason for the lack of effect of antipsychotic medications that do not affect cognitive symptoms (there is also the opposing view – [48]). Our research makes it possible to conclude that the early assessment of cognitive symptoms would enable the prediction of the course of the disease, the search for early therapeutic interventions and the discussion of additional strategies regarding cognitive disorders.

## Figures and Tables

**Figure 1 biomedicines-11-03114-f001:**
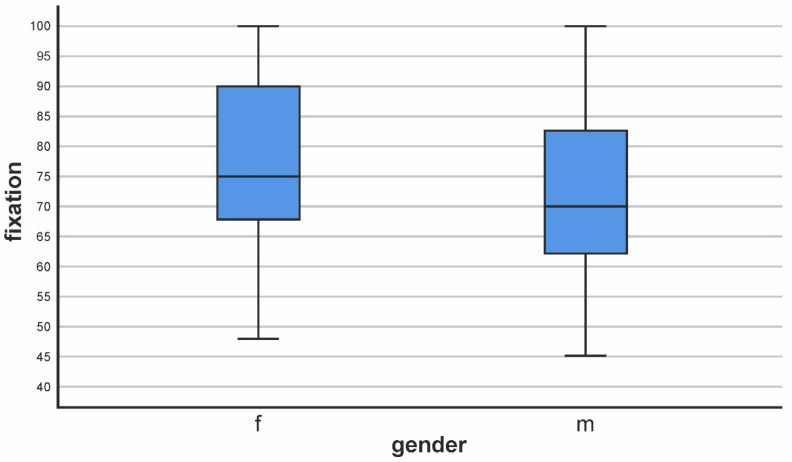
Distribution of male and female fixation values.

**Figure 2 biomedicines-11-03114-f002:**
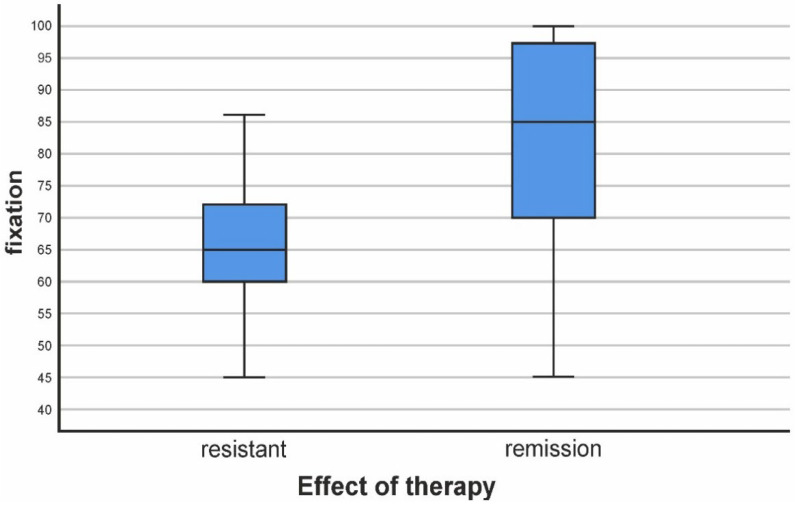
Relationship between fixation values in patients with resistance and those with treatment effect.

**Figure 3 biomedicines-11-03114-f003:**
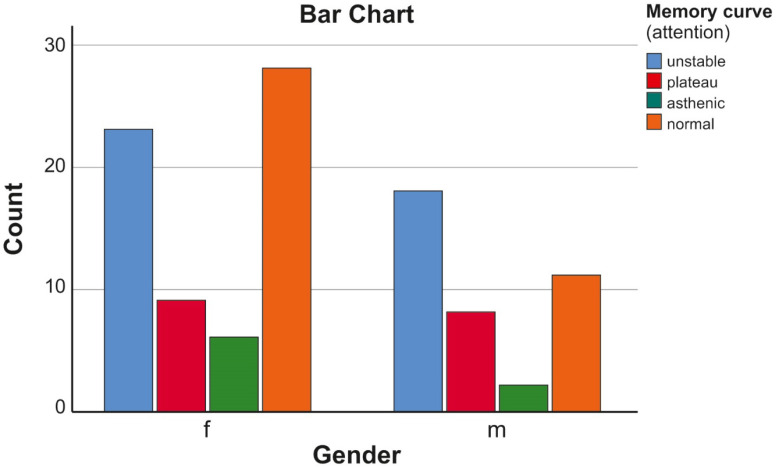
Relationship of the type of attention disturbances with the distribution by gender.

**Figure 4 biomedicines-11-03114-f004:**
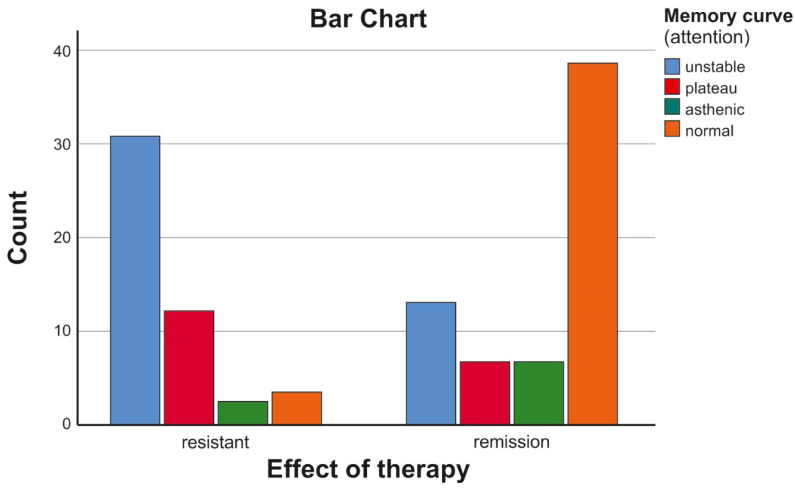
Distribution of the type of memory curve in patients with resistance and those with an effect of the therapy.

**Figure 5 biomedicines-11-03114-f005:**
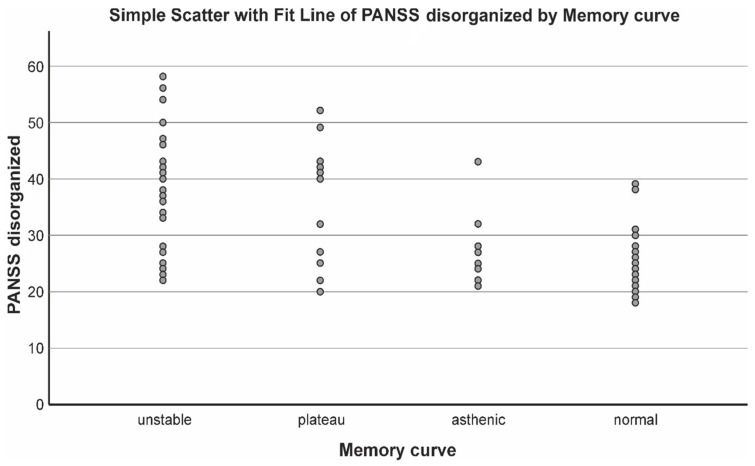
Relationship of attention disorders and PANSS—disorganized symptoms.

**Figure 6 biomedicines-11-03114-f006:**
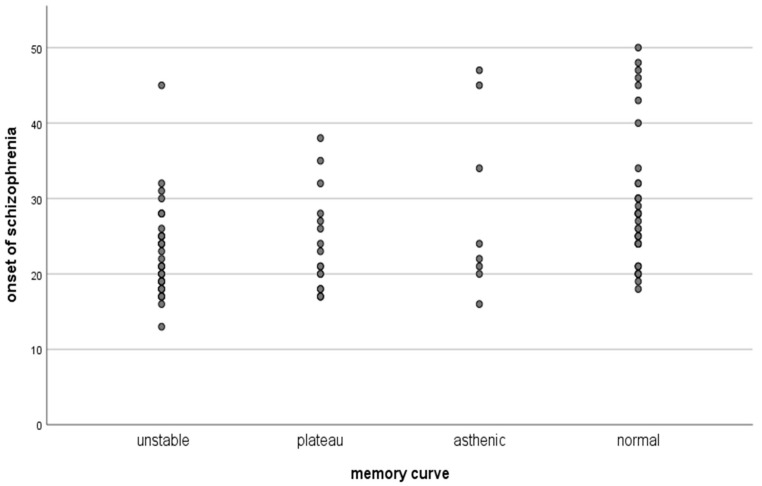
Relationship of attention disorders and the onset of schizophrenia.

**Table 1 biomedicines-11-03114-t001:** Distribution of patients’ age, age of onset of schizophrenia, duration of schizophrenia, BMI, height, education and handedness in both groups of patients.

Resistant SZ	Clinical Remission	Resistant SZ
Age (years)	36.98	37.25
Age of onset of SZ (years)	23.04	27.37
Duration of SZ (years)	14.31	9.87
BMI	26.6022	27.2217
Height	170.11	167.38
Education (years)	11.33	11.60
Handedness (right/left)	42/3	56/4
Sex (M/F)	20/25	19/41

**Table 2 biomedicines-11-03114-t002:** Distribution of male and female fixation values.

Fixation Score			
Gender	Mean	N	Std. Deviation
f	77,33	66	15,472
m	71,23	39	15,141
Total	75,07	105	15,561

**Table 3 biomedicines-11-03114-t003:** Presentation of statistical significance of differences between patients with resistance and those with therapy effect according to fixation value.

Mann–Whitney U	467,000
Asymp. Sig. (2-tailed)	0.000 (***)

**Table 4 biomedicines-11-03114-t004:** Relationship between fixation with the PANSS—disorganized symptoms, duration of schizophrenia and the level of dissociation.

	R2	β	t	*p* (Sig)
Step 1 PANSS disorganized	0.338	0.889	7.259	0.000 (***)
Step 2 PANSS disorganized Duration of Sch Step 3 PANSS disorganized Duration of Sch Dissociation score	0.388 0.411	0.350 0.130	6.865 2.880 4.869 2.646 1.988	0.000 (***) 0.000 (***)

**Table 5 biomedicines-11-03114-t005:** Assessment of the relationship between fixation disorders and the onset of schizophrenia process and its duration.

		Onset of the Sch	Duration of Sch
Fixation	Pearson Correlation	−0.274 **	−0.325 **
	Sig. (2-tailed)	0.005	0.001

**Table 6 biomedicines-11-03114-t006:** Relationship between reproduction and the PANSS—disorganized and PANSS—negative symptoms.

	R2	β	T	*p* (sig)
Step 1 PANSS disorganized	0.321	0.889	6.972	0.000 (***)
Step 2	0.351	0.350		0.000 (***)
PANSS disorganized			3.251	
PANSS negative			2.190	

**Table 7 biomedicines-11-03114-t007:** Relationship of reproduction with the duration and onset of schizophrenia process.

		Reproduction	Onset of the Sch	Duration of Sch
Reproduction	Pearson Correlation	1	0.153	−0.199 *
	Sig. (2-tailed)		0.120	0.041

**Table 8 biomedicines-11-03114-t008:** Relationship of the type of attention disturbances with the distribution by gender.

		Assessed Memory Curve (Attention)				
		Unstable	Plateau	Asthenic	Normal	Total
Gender	f	23	9	6	28	66
	m	18	8	2	11	39
Total		41	17	8	39	105

**Table 9 biomedicines-11-03114-t009:** Distribution of the type of memory curve in patients with resistance and those with an effect of the therapy.

		Assessed Memory Curve				
		Unstable	Plateau	Asthenic	Normal	Total
Treatment effect	resistant	29	11	2	3	45
	remission	12	6	6	36	60
Total		41	17	8	39	105

**Table 10 biomedicines-11-03114-t010:** Relationship between attention disorders and the PANSS—disorganized symptoms and the onset of schizophrenia.

	R2	β	t	*p* (Sig)
Step 1 PANSS disorganized	0.375	0.613	7.865	0.000 (***)
Step 2	0.445	0.561 0.268		0.000 (***)
PANSS disorganized			7.469	
Onset of Sch			3.587	

## Data Availability

The raw data supporting the conclusions of this article will be made available by the authors upon reasonable request.

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
