# Peer review of "Cognition in Patients with Schizophrenia: Interplay between Working Memory, Disorganized Symptoms, Dissociation, and the Onset and Duration of Psychosis, as Well as Resistance to Treatment"

_biomedicines, 2023, doi:10.3390/biomedicines11123114_

Round 1
Reviewer 1 Report (Previous Reviewer 2)
Comments and Suggestions for Authors
The Authors have responded in a satisfactory manner to all the Reviewers' queries and previous concerns were adequately addressed.
The manuscript has been substantially improved and is now much more clear and easy to read.
Author Response
Thank you very much for the evaluation and revision recommendations.
The Author
Reviewer 2 Report (Previous Reviewer 3)
Comments and Suggestions for Authors
I have enjoyed reading the revised manuscript entitled: Cognition in patients with schizophrenia. Interplay between working memory, disorganized symptoms, dissociation, the onset and the duration of psychosis as well as the resistance to treatment by Panov et al.
The authors responded and addressed the reviewer's comments. The revised manuscript might be acceptable.
I only have the following suggestions:
- There are spelling, punctuation and some grammar issues (e.g: lines: 66, 123….. etc). Sometimes the spaces between the words are missing, other times there are too many. This will apply to the whole manuscript.
- Please summarize the main theme of the article in a graphical abstract.
Comments on the Quality of English LanguageMinor editing of English language required.
Author Response
Thank you very much for the evaluation and revision recommendations.
Punctuation, spaces between the words and grammar were re-evaluated. Thank you very much for your comments.
A graphical abstract has been added according to your requirements.
The author
This manuscript is a resubmission of an earlier submission. The following is a list of the peer review reports and author responses from that submission.
Round 1
Reviewer 1 Report
Comments and Suggestions for Authors
The study conducted by Panov et al. explores the relationship between memory and attention and other clinical variables in schizophrenia patients. Specifically, differences between patients with resistant schizophrenia and those who are in clinical remission are studied. However, I believe the following concerns should be addressed before further consideration:
Main concerns
The objectives are vaguely operationalized, which makes it difficult to get an idea of what the authors aim to study. In this line, some variables such as sex do not appear to be relevant variables within the objectives of the study. However, separate analyses considering this variable are systematically carried out. The authors should explain why this is a core variable in the study.
Also, as this study includes longitudinal data, authors should use a statistical analysis appropriate for repeated measures. Mann-Whitney tests (check the spelling, line 214) are used for independent samples, so it is not clear whether authors analyze basal data or different time points. Maybe data is an average of the five years that patients were followed up. On the other hand, I would like to understand why having a sample of more than 100 patients, authors use non-parametric tests, when parametric tests seem to be more appropriate.
Weight, height, and body mass index (BMI) are extremely related -given that BMI is calculated based on height and weight-. Thus, there is a considerable risk of multicollinearity.
Usually, when the results of statistical analyses are given, the value of the statistic, as well as the degrees of freedom, should be given (for example, lines 249-250 or 284-285). Additionally, giving the same information in the text, in a table, and in a graph appears redundant (for example, paragraph “assessment of fixation”, Figure 1 and Table 2).
In the multiple regression analysis, I understand that differentiation between remission/no remission has not been included. I wonder why it was not included as a factor given that it seems a central variable in the study. Likewise, I would have included sex as a regressor instead of performing a separate analysis.
The results shown in Table 5 are totally unnecessary. If the onset of the disorder was not significant in the regression model when considering the rest of the factors, why do authors perform a correlation analysis? Similarly, if the duration of the illness was significant, what additional information provides a correlation with the dependent variable?
Some parts of the manuscript are written in Cyrillic: e.g., p. 9, lines 326-347; p. 10, line 381; p. 12, line 427; p. 21, lines 928-937…
Discussion: speaking about “differences without reaching statistical significance” (sentences 468-469 and 473-474) implies a misunderstood of the meaning of “statistical significance”. When results are not significant, it means that the variations found between groups (or whatever is being compared) are considered as due to random fluctuations of the sample values. Thus, the mean values of the groups should be considered “equal”. Therefore, expressions like “perform slightly worse” or “have lower scores” have no place in the discussion section under these circumstances.
Minor concerns
Abstract, line 21: “(DES)” is written twice.
Clinical contingent, line 167: What do authors mean by “Reduction of symptoms when assessed with the PANSS and BPRS scale by less than 20% for the observed period of time” as an inclusion criterion?
Clinical contingent, line 179: MMSE acronym is not explained.
Methods, lines 223-224: I would like to know how authors measured “the effect of the administration of the first antipsychotic medication”.
Figure 2 is missing.
Comments on the Quality of English LanguageEnglish should be revised.
Reviewer 2 Report
Comments and Suggestions for Authors
The present study is focused on a topic of scientifical and clinical relevance, and while it may lack striking elements of novelty, it does contain some elements of scientific interest.
However, the manuscript shows several very important quality issues that need to be thoroughly addressed before a more accurate review can be conducted.
General issues
-The use of English language is quite awkward in some sentences and several errors can be found through the manuscript. The manuscript should be submitted to professional proofreading or at least it should be checked by a native English speaker. Some sections are not presented in English language and should be appropriately translated are re-presented in the manuscript.
Introduction:
-While mentioning that cognitive impairment in people with schizophrenia responds poorly to currently available pharmacological treatments, the Authors should report that dedicated psychosocial interventions such as cognitive remediation are available and effective in treating this dimension (Vita A et al., Effectiveness, Core Elements, and Moderators of Response of Cognitive Remediation for Schizophrenia: A Systematic Review and Meta-analysis of Randomized Clinical Trials. JAMA Psychiatry. 2021;78(8):848-858. doi:10.1001/jamapsychiatry.2021.0620 and Lejeune JA et al., A Meta-analysis of Cognitive Remediation for Schizophrenia: Efficacy and the Role of Participant and Treatment Factors. Schizophr Bull. 2021;47(4):997-1006. doi:10.1093/schbul/sbab022). They should also mention and take into account the most recent international guidelines specifically dedicated to the treatment of cognitive impairment in schizophrenia (Vita A et al., European Psychiatric Association guidance on treatment of cognitive impairment in schizophrenia. Eur Psychiatry. 2022;65(1):e57. doi:10.1192/j.eurpsy.2022.2315).
-The aims of the study and the main research hypotheses should be explicitly mentioned at the end of the Introduction section in a dedicated paragraph.
Methods:
-The “Clinical contingent and methods” section should be renamed as “Methods”, the “Clinical contingent” subsection should be renamed as “Participants” and the “Methods” section should be split into two subsections, “Measures” and “Statistical Analyses”. The statistical analyses adopted in the study should be thoroughly described in the appropriate section.
-More details should be provided regarding the process recruiting participants, including dates of start and conclusion of the recruitment period and the exact location where the study was conducted. This is fundamental to improve the clarity of the work and to allow reproducibility of results.
-Information and details (such as date and code of approval) regarding the approval of the Local Ethical Committee or Institutional Review Board to conduct the present study should be included in the manuscript and reported in the Methods section.
All these changes are fundamental to assess the reproducibility and validity of the study results and the discussion of the presented data.
Comments on the Quality of English LanguageThe use of English language is quite awkward in some sentences and several errors can be found through the manuscript. The manuscript should be submitted to professional proofreading or at least it should be checked by a native English speaker. Some sections are not presented in English language and should be appropriately translated are re-presented in the manuscript.
Reviewer 3 Report
Comments and Suggestions for Authors
The authors focused on: Cognition in patients with schizophrenia. Interplay between working memory, disorganized symptoms, dissociation, the onset and the duration of psychosis as well as the resistance to treatment. Interesting topic and good Results part, offering the perspective of a promising paper. However, I have the following suggestions related to the improvements that should be added:
You are encouraged to use the journal template to prepare the manuscript.
It is advisable to develop future research directions based on the identified unmet needs in order to solve the limitations of the present study.
I consider that it needs substantial additions in the sections of material and methods and more appropriately correlated with results and discussion sections
The References list contains several incomplete information on certain bibliographic resources, generally the lack of inclusion of pages or at the level of including authors. I suggest completing them.
It is mandatory to check and correct the numbering of section” References” because there are numbering inadvertencies.
Comments on the Quality of English LanguageI recommend to the authors that the entire manuscript be written in English. Please check the paragraphs between the lines 326-346, 381 and 427.
The English language, although quite good, should be improved.